# Distance and Intensity Profiles in Division I Women’s Soccer Matches across a Competitive Season

**DOI:** 10.3390/sports9050063

**Published:** 2021-05-12

**Authors:** Mario Norberto Sevilio de Oliveira Junior, Christiano Eduardo Veneroso, Guilherme Passos Ramos, Kelly E. Johnson, Justin P. Guilkey, Alyson Felipe da Costa Sena, Christian Emmanuel Torres Cabido, Jason M. Cholewa

**Affiliations:** 1Department of Kinesiology, Coastal Carolina University, Conway, SC 29526, USA; kjohns10@coastal.edu (K.E.J.); jguilkey@coastal.edu (J.P.G.); 2Department of Physical Education, Federal University of Maranhão—UFMA, São Luis 65080-805, MA, Brazil; cveneroso@hotmail.com (C.E.V.); afdcs94@hotmail.com (A.F.d.C.S.); christianemmanuel@gmail.com (C.E.T.C.); 3Brazilian National Football Confederation (CBF), Rio de Janeiro 22775-055, RJ, Brazil; guilherme.passos@cbf.com.br; 4Department of Exercise Physiology, College of Health Sciences, University of Lynchburg, Lynchburg, VA 24501, USA; Jason.m.cholewa@gmail.com

**Keywords:** football, team sport, physical demands, monitoring, competition

## Abstract

Women’s participation in soccer has increased rapidly. The purpose of this study was to evaluate the physiological demands imposed on women’s NCAA Division I soccer players across a season according to speed, total distance traveled, and numbers of sprints measured via GPS (Polar Team Pro^®^). Eighteen athletes (19.2 ± 1.1 years, 50.5 ± 6.5 mL/kg/min and 23.4 ± 3.6% fat) participated in this study. The analysis was obtained through 13 official matches. Speed zones were defined as Zone 1 (1.0 to 5.99 km·h^−1^), Zone 2 (6.0 to 10.99 km·h^−1^), Zone 3 (11.0 to 15.49 km·h^−1^), Zone 4 (15.5 to 19.9 km·h^−1^) and Zone 5 (sprint > 20 km·h^−1^), with Zones 4–5 considered as high intensity running. Individual differences in playing time and total distance were highly variable due to substitutions. Average distance traveled per game in quartiles was 3.9 km, 5.6 km and 7.4 km in the 25th, 50th, and 75th quartiles, respectively. Relative to playing time, players travelled an average of 113.64 ± 17.12 m/min (range: 93.7 to 143.5 m/min) and ran one sprint every 4.12 ± 2.06 min. When distance was summated and analyzed for the entire team, significant difference between halves was found for speed Zones 2, 3 and 4, with greater values found in the first half. Total distance, high intensity running and sprint distance were significantly less than previously recorded in professional players. These findings suggest that coaches should consider the unique physiological demands and recovery opportunities present in NCAA play when constructing practices and conditioning programs.

## 1. Introduction

Women’s participation in soccer has increased rapidly since the first Women’s FIFA World Cup was held in 1991 [1], and soccer is currently considered one of the most popular sports for women’s participation in the United States. Title IX of the Education Amendment of 1972 prohibits sex discrimination in any education program or activity receiving federal financial assistance and has opened more opportunities for women to compete at the high school and collegiate level [2]. As a consequence, the number of athletes and the level of competition in National Collegiate Athletic Association (NCAA) soccer has increased in recent decades. As a result, it is likely that collegiate soccer players are being exposed to greater training volumes and competition demands than previously estimated [3].

Soccer requires high levels of both anaerobic and aerobic fitness which varies by the level of competition, position, and match style of play [4]. Recently, the use of global positioning systems (GPS) has gained popularity among sport scientists and coaches because they provide direct measurements of total distances traveled. Additionally, GPS can break down the distances covered by speed categories (intensity zones) [3,5,6]. At the national and professional levels, the running demands in both men and women’s soccer athletes have been extensively studied [7,8,9]. For professional women soccer players, single sprint distances (15.1 ± 9.4 m), sprint times (2.3 ± 1.5 s), and rest between sprints (2.5 ± 2.5 min) have been reported [10].

NCAA rosters are larger than professional rosters and NCAA rules allow for unlimited substitutions, so match play is different. Therefore, designing practice and strength and conditioning programs based off professional and international data may be inappropriate. To our knowledge, four studies have documented external workloads of NCAA women’s soccer players over a competitive season. In the first study, Wells et al. [11] reported data for only 11 out of 20 athletes that played for 55 min (~61% of each game) or more each game, whereas 19 of the 20 players played in all 21 games (regular and post season), demonstrating the high variability in game demands due to larger rosters and unlimited substitutions compared to professional rules. In the Wells et al. [11] study, total distance covered was 7481 ± 958 m with sprinting accounting for 86 ± 80 m and high intensity running (>13 km/h) accounting for 1412 ± 244 m. In the second study, Gentles et al. [3] analyzed 17 matches for all 22 players on the roster and reported an average total distance of 5480 ± 235 m with 590 ± 35 m of the total distance covered at velocities over 15 km/h. The large discrepancy in total distance was likely the result of including all players in the analysis, which reduced average playing time to 45 ± 26 min [3]. In the third study, McFadden et al. [12] reported data for 9 players who played at least 45 min per game during a competitive season as well as for all players (*n* = 16) who were substituted into at least 50% of the 23 matches. In this study, the total distance and number of sprints per game were 8.31 ± 0.9 km and 14 ± 5 sprints per game for athletes that played at least 45 min per game, and 6.06 ± 2.7 km and 11 ± 5 sprints for those who played in at least 50% of the matches. The fourth study only analyzed athletes who played a minimum of 45 min in at least 50% of the games (*n* = 11). Bozzini et al. [13] reported athletes covered an average of approximately 104 m/min and performed 0.16 sprints per minute (approximately 1 sprint every 6 min). These large discrepancies in total distances covered per game between studies were likely due to differences in final analysis inclusion criteria. For example, Gentles et al. included all players (*n* = 22) and reported a large variance in playing time (45 ± 26 min) compared to much smaller variances in playing time reported by Bozzini et al. (*n* = 11, out of conference games 79.7 ± 17.3 min and 89.9 ± 15.3 min for in conference games).

Understanding the competitive demands during NCAA matches in Division I women’s soccer is necessary to guide training methods for maximizing physical development, reducing the risk of injury, determining athlete readiness, and implementing athlete recovery methods. Thus, the purpose of this study was to evaluate the physiological demands imposed on women NCAA Division I soccer players during the games of a university season according to speed, total distance covered (TDC) and numbers of sprints measured via GPS.

## 2. Methods

Eighteen athletes (19.2 ± 1.1 years, 167.9 ± 2.24 cm, 64.6 ± 7.8 kg, 50.5 ± 6.5 mL/kg/min and 23.4 ± 3.6% fat) from a NCAA Division I university participated in this study. All subjects gave written informed consent for inclusion before they participated in the study. The study was approved by the Institutional Review Board of Coastal Carolina University, Conway, South Carolina (#2018.140) and all the athletes signed a consent form to participate in the study.

All data was collected during the second half of 2018, which represents the completion of an NCAA Women’s Division I Soccer season. The present study was a descriptive and observational design and conducted to characterize and analyze the game activity profile of a South Carolina University playing in the Sun Belt Conference. In total, 13 matches were recorded out of a total of 17 matches. It should be noted that it was not possible to record the first match of the season because the GPS was not yet available. Three other matches were not recorded during September because Hurricane Florence resulted in state government-mandated evacuations, and the closure of campus for three weeks. This forced the team to relocate training and matches outside of the state of South Carolina.

## 3. Methodological Procedures

Subjects reported to the Exercise Physiology Laboratory before the start of the soccer season to complete a body composition and maximal oxygen uptake (VO_2_peak) measurement. Prior to arrival, subjects were instructed to refrain from alcohol intake for 48 h and maintain normal hydration prior to all testing. Body composition was measured via the Bod Pod (Cosmed^®^, Rome, Italy) according to manufacturer protocols with a predicted lung volume. Following the body composition measurement, subjects completed a self-selected dynamic warm up and then a graded exercise test to maximal exertion was performed in order to measure VO_2_peak. The protocol was a soccer specific graded exercise test on the treadmill using a spirometer with an open gas analyzer (Trueone 2400 Parvo Medics^®^, Parvo, UT, USA). The speed-based protocol used was designed with stages that were MET-equated to the Bruce protocol and has been described and validated in soccer players [14]. In brief, throughout the protocol every two minutes the speed increased, but the incline remained constant until subjects reached volitional exhaustion. The speeds for each stage were 6.43, 7.88, 9.97, 11.74, 13.67, 15.61, 17.06, 18.18, 19.79, and 21.08 km/h at a constant 2% incline.

The analysis of game movements was obtained through 13 official matches. All players that participated in games were included in the analysis. The GPS analyses were linked before heating to allow the devices to locate the required satellites and the units were placed at the height of the thorax (mediastinum line) before each match.

## 4. Match Analysis

The speed zones, number of sprints, and total distance covered were obtained through the use of a Polar-branded GPS (Polar Team Pro^®^ model, which features a 10 Hz integrated GPS and a 200 Hz MEMS motion sensor. Polar Electro, Kempele, Finland). The units were mounted on the front of each player’s chest using an adjustable Neoprene strap and attached 60 min before the start of each match. This procedure allowed the connection and acquisition of 12 satellites and synchronization with the software. In addition, the same unit was used by each player in all matches to reduce inter-unit measurement error. The following categories were used to classify speed zones over distances covered by athletes: Zone 1 (1.0 to 5.99 km/h), Zone 2 (6.0 to 10.99 km/h), Zone 3 (11.0 to 15.49 km/h), Zone 4 (15.5 to 19.9 km/h) and Zone 5 (>20 km/h) [8]. In addition, Zone 4 was classified as the High-Speed Zone (>15.5 km/h) and Zone 5 as the Maximum Speed/Sprint Zone (>20.0 km/h). From this classification, the total number of sprints in the first and second halves was quantified. Additionally, the distance (m) covered in every minute of the game was presented as a total for the team and for each athlete for each of the 13 games during the season. Finally, the average sprint frequency for each athlete in the matches was calculated.

Previously conducted studies on 10 Hz GPS devices have shown that this GPS device has good validity and reliability (5% CV) for total distance, linear running, and change of direction [15,16]. For the Polar Team Pro®️ model used in this study, the ICCs have been reported as extremely high for running less than 13.99 km/h and 14–19.99 km/h (0.99 and 0.92, respectively), and high for total distance and sprinting (>20 km/h) (0.63 and 0.65, respectively) [17]. Additionally, there does not appear to be a benefit of using a 15-Hz GPS over a-10 Hz GPS device with regard to linear and team sport simulated running [16].

## 5. Statistical Analysis

Data are presented as means and standard deviations. Comparisons of the measurements of the first and second half of games were made through two-way repeated measure ANOVA with a post hoc Student’s *t*-test, when necessary. All statistical analyses were performed using statistical software SPSS (SPSS version 18; IBM, Armonk, NY, USA) with statistical significance being set at *p* < 0.05.

## 6. Results

The cumulative distance covered by all players (team distance) was 96.39 km. In two matches the team achieved distances greater than 100 km (104.99 and 101.19 km), and in two matches less than 90 km (89.70 and 88.87 km). Figure 1 displays the team’s total number of sprints according to the first and second half. The mean values found were 128 and 125 sprints in the first and second half, respectively, with no significant differences occurring (*p* = 0.74). The team achieved the highest performances in the following matches: 13° (153 sprints), 2° (152 sprints) and 12° (150 sprints), all being obtained in the second half. On the other hand, the lowest performances (93, 100 and 105 sprints) occurred in the second half, in the respective matches: 9°, 10°, 5°, as seen in Figure 1.

The values shown in Figure 2 are divided into high intensity Zones (4 and 5) and low intensity Zones (1 to 3), and represent the cumulative distances covered by all players. Significant differences were found in Zones 2, 3 and 4. Specifically the first half contained more distance within Zones 2, 3, and 4 than the second half. In Zones 1 and 5, there was no significant difference between the first and second half (*p* ≥ 0.60). On the other hand, the distance covered in Zones 3, 4, and 5 was significantly different, a result expected as a function of the velocities that make up each zone.

Table 1 displays the total distances (km) covered by the athletes distributed in percentiles, based on the greatest and least distance between players. Due to the large number of substitutions that occurred in the matches we observed, some athletes who participated only 5 min in a single match covered 720 m, and athletes who played the full 90 min of the match reached distances greater than 10.1 km. The average distance covered in the 25th percentile was over 3.934 km, the 50th percentile was over 5.692 km and, finally, in the 75th percentile, greater than 7.449 km.

Table 2 displays the mean distance covered per game and distance covered relative to playing time (m/min) for each speed zone.

Figure 3 illustrates the distance covered relative to minutes played by each athlete for all 13 matches. Table 3 displays the distance covered relative to minutes played in quartiles for all 13 matches. Large, but not statistically significant, inter-individual differences between players were found, with the 16th and 20th players covering 143.45 ± 44.01 and 140.30 ± 26.13 m/min, respectively, and the 3rd player presenting the lowest distance of 93.67 ± 12.68 m/min. The mean value of the athletes during all matches was 113.64 ± 17.12 m/min.

As soccer is characterized by short duration and high intensity actions such as sprints, Figure 4 displays the frequency of sprints as the duration of time between each sprint. Athletes performed one sprint every 2 to 4 min, on average. The overall team average was approximately one sprint every 4.16 ± 2.11 min in the first half and 4.08 ± 2.00 min between sprints in the second half; the differences were not significant (*p* = 0.47).

## 7. Discussion

The purpose of this study was to evaluate the demands of a competitive soccer season in women NCAA Division I soccer players using zones of speed, total distance covered (TDC) and numbers of sprints. The average of TDC by the team in this study was 96.39 km, which is similar to those found in professional teams such as the Danish first division (103 km) [4], the Brazilian U20 team (88 km) [8] and the Australian professional league (97 km) [18]. However, the TDC per player (5841.87 m) and frequency of sprints (one sprint approximately every 4 min) was significantly less than that covered by professional women players (8500 to 10,300 m and one sprint every 2.5 min, respectively) [16].

The average TDC per player in this study (5841.87 m) and in the study by Gentles et al. [3] (5480 m) were, respectively, 21.92% and 26.8% less than in Wells et al.’s study [11] (7481.9 m), and 30% and 34% less than in McFadden et al.’s study [13], respectively, despite all four studies analyzing NCAA Division 1 women’s soccer games. Total high intensity running distances (>15 km/h) were also similar between our study (609 ± 151 M) and Gentles et al.’s study [3] (590 m) but less than in Wells et al.’s study [11] (1412 m). It is important to note that Wells et al. [11] defined high intensity as greater than 13 km/h, which could be a reason for the two-fold differences in distances of high-speed running. Additionally, discrepancies in match play locomotion are due to differences in reporting the data, as we reported for all players whereas Wells et al. [11] and McFadden et al. [13] only reported for players competing in greater than 55 and 45 min per game, respectively. As a result of including all players, large interindividual differences were found for minutes played in this study (43.1 ± 28.1, range 5 to 90 min), but were similar to those reported by Gentles et al. [3] (45.3 ± 26 min). However, playing time in the present study was considerably less than Wells et al. [11] (72.6 ± 12.9 min) or Bozzini et al. [12] (approximately 85 ± 16 min). Given the large differences in play time, it may be appropriate to examine distances covered relative to playing time. Total distance relative to playing time in the present study (113.6 m/min) was similar to Wells et al. [11] (104.5 m/min) and Bozzini et al. [12] (104 m/min).

Although the speed bands were defined different between studies (i.e., Wells et al. [11] defined Zone 4 as 15.96 to 21.9 km/h and the present study and Bozzini et al. [12] define it as 15.5 to 19.9 km/h; Wells et al. [11] defined sprints as >22 km/h and we and Bozzini et al. [12] defined sprints as >20 km/h), some similarities can be noted. For example, running in Zone 4 players covered 7.93 ± 2.47 m/min, ~10.15 ± 2.5 m/min, and 9.8 ± 5.6 m/min in the study by Wells et al. [11], Bozzini et al. [12] and the present study, respectively. On the other hand, players covered 1.22 ± 1.17 m/min, 3.13 ± 2.0 m/min and 3.9 ± 3.1 m/min sprinting in the study by Wells et al. [11], Bozzini et al. [12] and the present study, respectively. These differences in sprinting distances are likely due to differences in the definition of sprinting between the three studies. In addition to the differences in the definition of the various zones, differences in sprinting distance may also be due to fatigue as a result of longer playing times or a strategic conscious conservation among starting players. While we cannot speculate on which mechanism is responsible, we found an inverse correlation between playing time and relative Zone 5 (*r* = −0.44) and Zone 4 (*r* = −0.46) distance in this study, which may support these hypotheses.

Although the average total team distance in this study was similar to those reported in elite women soccer players, individual average total distance differences were 40–50% less than those reported in elite players [19]. This is likely due to differences in substitution rules because competitions at the professional level allows a maximum of three substitutions per game, whereas in the NCAA the substitutions are unlimited, and allow the same athlete to enter several times in the same game, increasing the time in Zones 4 and 5 [3,11,12,13]. The NCAA rule is used because matches take place on weekends and often with a recovery time between matches of only 36 h.

High intensity running in elite women soccer players has been reported to average between 1680 and 2407 m. A reduction of approximately 6.5 to 10% in high intensity running has been observed during the second half of the game compared to the first half, with the greatest reductions occurring in the final ten minutes of the game [10,20]. In our study, players averaged 609 m of high intensity running, with a 5% reduction occurring between the first half and the second half. This reduction was the result of 9.3% less distance in Zone 4, which is a greater reduction than that found in the Brazilian sub 20 soccer (7.52% reduction between first and second halves) [8]. The running distances within Zone 4 are greater than those found within in Zone 5. This could result in using the glycolytic system of energy production and, consequently the effects of reductions in muscle glycogen later in the game and the accumulation of hydrogen ions as game play continues [21]. On the other hand, Zone 5 distances increased by 6.9% from the first to second half. In our study, an average of 16.5 players participated in each match, and 19.3 substitutions were used. This peculiar characteristic likely contributed to the matches having a significant percentage of high intensity running and at the same time maintaining this intensity in both the first and the second half. Substitutions allowed athletes a greater recovery during the competition, as the actions performed in Zone 5 present durations close to 2 s, relying predominantly the phosphagen system, whereby creatine phosphate resynthesis can occur by approximately 90% in up to 1 min [22,23].

The following limitations should be considered when interpreting the results of this study. First, we were unable to express the results based upon positions played, as individual athletes were often required to play multiple positions in a single match (i.e., mid-fielder and forwards). Second, unlimited substitutions leading to large inter-individual variances reduce the ability to detect statistical differences. To account for this limitation, we report the data as both team totals and relative to minutes played. Third, we acknowledge that GPS becomes less sensitive with increased acceleration and change of direction, and distances covered in Zone 5 may present with errors. Finally, to the best of our knowledge, direct comparison data of the Polar Team Pro to the GPS devices used in studies cited in this paper (i.e., Zephyr and Catapult) are not available. Therefore, future research is necessary to quantify these differences as to better compare total distance and high intensity running between various levels of women’s NCAA soccer.

## 8. Conclusions

Our goal was to quantify the running demands of an NCAA Division I women’s soccer team during a competitive season. Our primary findings were that total distance, high intensity running, and sprint distance was significantly less than previously recorded in professional women’s soccer play. There were no significant differences for total distance, sprints and Zone 5 for the first half in relation to the second half. These differences could be the result of unlimited substitutions in NCAA play, while professional soccer allows for only three substitutions per game. As such, coaches and strength and conditioning professionals should consider the unique physiological demands and recovery opportunities present in NCAA play when constructing practices and conditioning programs.

## Figures and Tables

**Figure 1 sports-09-00063-f001:**
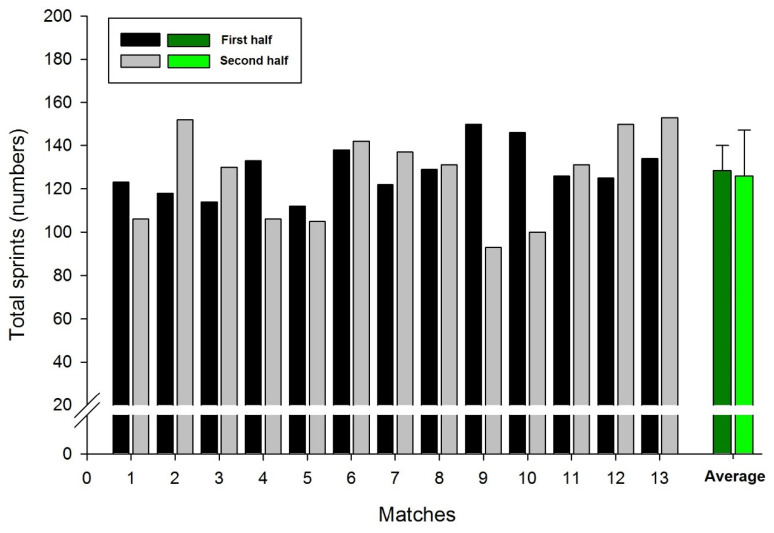
Team’s total number of sprints according to the first and second half.

**Figure 2 sports-09-00063-f002:**
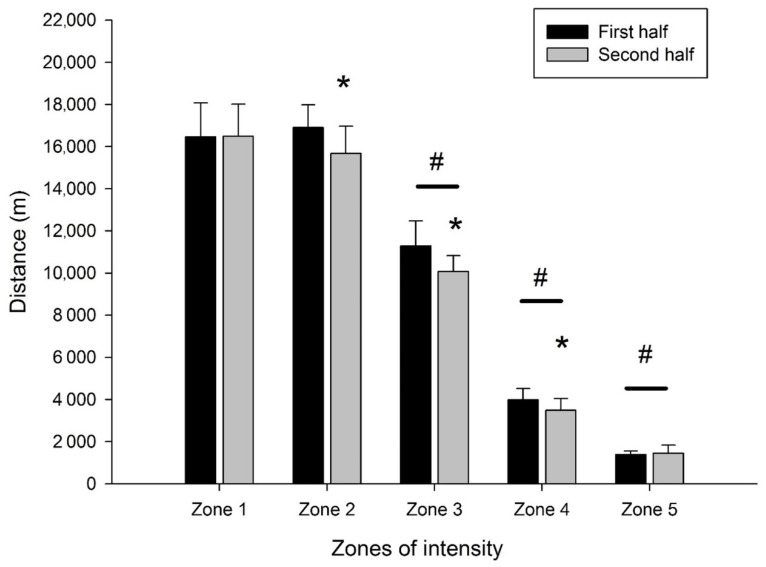
Cumulative distances covered in low (1–3) intensity zones and high (4–5) intensity zones. ***** *p* < 0.05 significantly difference of first half; **#** *p* < 0.05 significantly difference between zones.

**Figure 3 sports-09-00063-f003:**
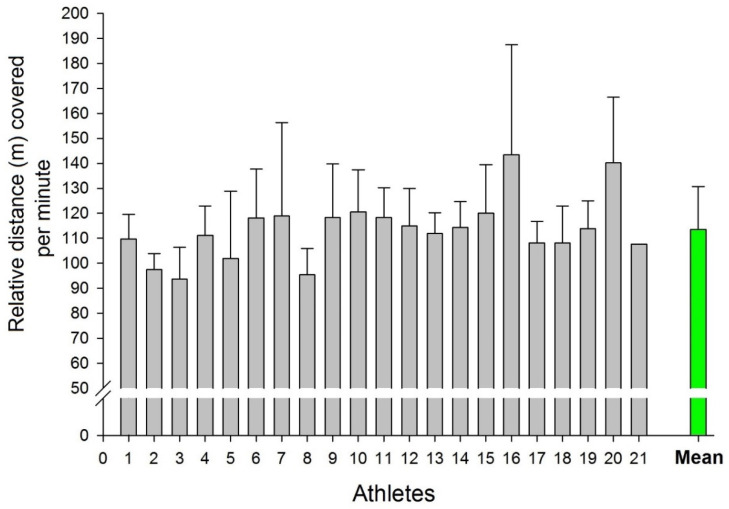
Relative distance covered per minute of playing by each athlete for all 13 matches.

**Figure 4 sports-09-00063-f004:**
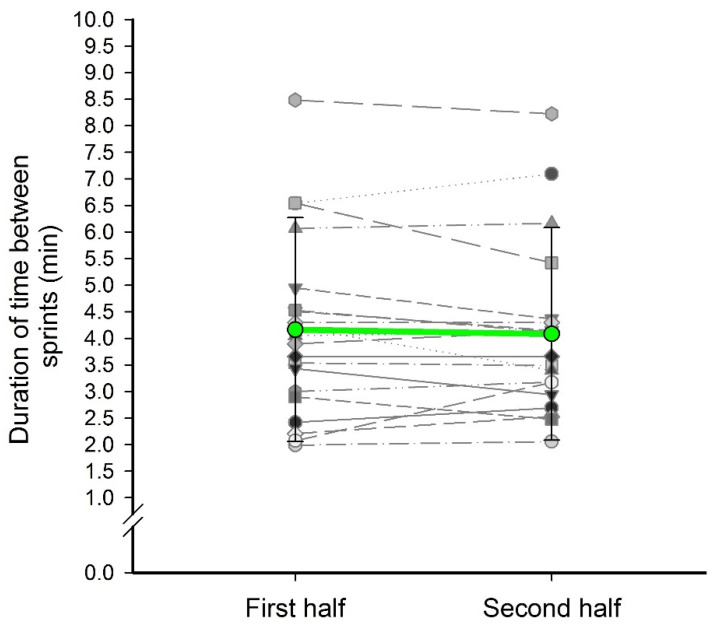
Frequency and difference between the first and second half of sprints as the duration of time between each sprint.

**Table 1 sports-09-00063-t001:** Total distances (km) covered by the athletes distributed in percentiles.

Matches	Percentile 25	Percentile 50	Percentile 75
1	4.223	5.900	7.577
2	3.571	5.227	6.883
3	2.936	5.151	7.366
4	3.156	5.195	7.234
5	4.520	5.974	7.428
6	3.772	5.534	7.296
7	4.199	5.708	7.217
8	3.396	5.205	7.014
9	4.026	5.766	7.506
10	4.940	6.225	7.510
11	3.730	5.653	7.576
12	5.042	6.757	8.472
13	3.642	5.702	7.762
Average	3.934	5.692	7.449

**Table 2 sports-09-00063-t002:** Mean (km) and relative distances (m/min) for each speed zone.

Zone	Distance (km)	Relative Distance (m/min)
1	1.924 ± 0.691	42.7 ± 21.1
2	1.913 ± 0.720	41.7 ± 17.4
3	1.253 ± 0.520	21.2 ± 11.7
4	0.434 ± 0.180	9.8 ± 5.6
5	0.167 ± 0.099	3.9 ± 3.1

**Table 3 sports-09-00063-t003:** Relative distance covered per minute of playing time (m/min) by the athletes distributed in percentiles.

Matches	Percentile 25	Percentile 50	Percentile 75
1	99.8	110.7	118.0
2	103.3	111.8	115.5
3	102.2	117.9	129.7
4	94.5	122.8	140.8
5	91.4	115.9	136.9
6	98.3	113.5	133.9
7	103.5	114.4	122.5
8	95.7	107.2	122.2
9	103.5	114.4	122.5
10	92.3	115.1	131.7
11	89.6	102.3	116.9
12	97.8	107.7	131.6
13	100.1	135.8	170.5
Average	108.1	116.9	128.3

## Data Availability

Data sharing is not applicable to this article.

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
