# Peer review of "Distance and Intensity Profiles in Division I Women’s Soccer Matches across a Competitive Season"

_sports, 2021, doi:10.3390/sports9050063_

Round 1
Reviewer 1 Report
The authors have carefully analyzed speed and distance of movement during elite women's soccer. A number of details are important to specify for the readers to appropriately interpret the results.
First, how was " sprint" Explicitly defined? Presumably, sprint is not rapid running from a stationary start, but acceleration for a short period of time at a variable rapid running rate, which is unlikely to be At a constant velocity.
How accurate are the GPS monitors to identify sprints of short distance, for example 25 m or less?
There is a consistent reference in mini figures and in text results to distance expressed as a metres per minute. Metres per minute is of course velocity and not distance. This Variable needs to be clarified.
The authors need to make much more clear the interest in and the information contained in total distance and distribution of velocities for the team as a whole, as opposed to for individual players. Given the variable number of players and three minutes played per player, I am Not sure what the academic value is to measure the overall distances and velocities for the team as a whole, except as background information. What is presumably much more interesting and useful is the inter-individual variability, and distribution, of distances traveled, distribution of velocities in the various velocity groupings, and the number of sprints per athlete per game. Some of this information seems to be in table four, but is not sufficiently detailed for useful analysis. The units on the Y axis in figure 4 to not make any sense, and needs to be modified
Reviewer 2 Report
The present study aimed to evaluate the physiological demands imposed on female NCAA I soccer players during 13 official matches of a season using zones of speed, total distance traveled, and numbers of sprints measured via GPS. Eighteen athletes participated in this study. The team’s average total distance covered was 96.39 km. The mean values found were 128 and 125 sprints in the first and second half, with no significant differences between halves. In the zones 2, 3 and 4, the first half had values higher than those found in the second half. Moreover, total distance, high intensity running and sprint distance was significantly less than previously recorded in professional players.
Overall, the manuscript is interesting and is well written. However, the authors could better add more details, what may help the reader to better understand the findings. In this sense, minor revisions are suggested:
Abstract
For this reviewer, the relative total distance (m/min) and differences between halves should be showed in abstract section.
In the Introduction section
The authors have to think about an international reader, who perhaps doesn´t know what supposed Title IX of the Education Amendments Act of 1972 in United States, which ensures women the same rights as men regarding school and university sports programmes. Please, explain something more in those lines.
In the Materials and Methods section
To analyze the match running profile, distance traveled over full game is an important data. However, the authors may consider further splitting up this data (e.g. successive period of 15 min of match play). If the authors have these data, please provide.
On the other hand, device used in this study allows to know heart rates (HRs) and therefore the relationship between external and internal load (index of performance efficiency -effindex-) describes in previous studies (Suárez-Arrones et al, 2015, PMID: 25289717; Torreño et al, 2016, PMID: 26816391). In this sense, it would be interesting to know the average HR recorded expressed as % maximal HR.
At the same time, it would be interesting to know which playing position displayed the highest and fittest levels of physical and physiological demands. Unfortunately, these data are no provide in this study because changes on players’ positions across the season.
Reviewer 3 Report
Thank you for your submission to Sports. It is great to see a study on women's collegiate soccer.
In general, the manuscript is well-written paper with nice visuals and presentation of the data.
What follows are some suggestions designed to improve the quality of the paper.
General comments
Since the topic pertains to women’s collegiate soccer not female soccer, please change "female" to "women" throughout the manuscript. Also, avoid writing in the first person (e.g., “I” or “we”), and change throughout the manuscript. When listing values the use of a zero placeholder needs to be consistent. For example, some places it is listed as 0.60 and others as .46.
Introduction
Last paragraph. Both sentences begin with "Therefore". Select another linking word.
Methods
Were subjects consented prior to data collection?
Provide more detail on BodPod and VO2 testing procedures. For example, what instructions were provided to subjects? When was testing conducted?
Polar – was the system calibrated prior to use? Did athletes have specific units assigned?
Results
Team data are always interesting but don’t tell us much about the individual response or position group response. It is suggested the authors consider data stratification. For example, might the authors divide players into starters/non-starters or classify based on number of minutes played? Are those players that are sprinting more in the second half playing less time in the first half so they are fresher? Are those players more aerobically fit when we look at their VO2? This is mentioned in the discussion as a limitation, but perhaps going back and classifying <45 or >45 minutes.
The authors are referred to this reference as an example:
Bozzini, Brittany N.; McFadden, Bridget A.; Walker, Alan J.Varying Demands and Quality of Play Between In-Conference and Out-of-Conference Games in Division I Collegiate Women's Soccer Journal of Strength and Conditioning Research. 34(12):3364-3368, December 2020.
In addition to playing time, the authors might look at this data by position, after all it is a descriptive paper so crazy stats don't need to be done on the small sample sizes. However, can look at number of sprints, total distance etc. with respect to playing position. Which positions are running more/sprinting more? And that would tie into the discussion of how unique periodization is needed from a position standpoint. Something to discuss and potential in variation and individualization.
Table 1
Clearly state how the percentiles for total distance there determined, from what value were they based?
Figure 4
Seeing these data as percentile groups instead of averages would be of greater interest.
Discussion
This section needs to be strengthened with additional, relevant citations from the literature. When citing references please don’t use the citation number in place of the author’s name.
The interesting story with this study is the individual response not the team response. What does it tell the reader if the average team distance is equal to elite women athletes but the individual differences are almost 50% less? Consider reframing your data (as suggested in the Results above) to tell a more interesting story.
Author Contributions
Please explain, using complete initials, what the role was of each author. The journal requires an extensive list in this section. Refer to their guidelines for assistance.
IRB
Include the approval number and date of approval in this section.
Informed Consent Statement
Please include this information in the Methods section
References:
The review of literature should be strengthened, especially on the topic of load monitoring in soccer. There are several papers missing that could be used to strengthen the current manuscript. Therefore, it is suggested the authors dig deeper into the literature. A couple of examples are listed below.
McFadden, BA, Walker, AJ, Bozzini, BN, Sanders, DJ, and Arent, SM. Comparison of internal and external training loads in male and female collegiate soccer players during practices vs. games. J Strength Cond Res 34(4): 969–974, 2020
Bozzini, Brittany N.; McFadden, Bridget A.; Walker, Alan J.Varying Demands and Quality of Play Between In-Conference and Out-of-Conference Games in Division I Collegiate Women's Soccer Journal of Strength and Conditioning Research. 34(12):3364-3368, December 2020.
Round 2
Reviewer 3 Report
General comments:
Thank you for the revised manuscript. Clearly, the authors worked hard to improve the paper's quality. However, there are many grammatical errors in the revised version. It is recommended that the authors seek editing assistance from a native English speaker in order to improve the quality, flow and readability of the paper. It is suggested that the authors do a search and replace throughout the manuscript in order to change "female" to "women".
Specific comments:
Methodological Procedures - The authors state VO2max test was performed until volitional fatigue. Wouldn't this be VO2 peak? Please rephrase. Also, in the methods where the VO2 information is included, VO2 needs to be added to the parentheses.
Author Response
Thank you for your comments. We have used the search feature in MS Word to change all instances of female to to women or women's. The only occurrence of female is now in the titles of cited studies in the references.
We have changed VO2max to VO2peak, and used parenthesis as indicated.
Two Native-English speaking co-authors have proof read and edited the study.